# Curriculum Learning for Vision-and-Language Navigation

**Jiwen Zhang**[1], **Zhongyu Wei**[1,2]*, **Jianqing Fan**[1,3], **Jiajie Peng**[2]
[1]School of Data Science, Fudan University, China
[2]Research Institute of Intelligent and Complex Systems, Fudan University, China
[3]Department of Operations Research and Financial Engineering, Princeton University, USA
{jwzhang16,zywei}@fudan.edu.cn, jqfan@princeton.edu, jiajiepeng@nwpu.edu.cn

## Abstract

Vision-and-Language Navigation (VLN) is a task where an agent navigates in an embodied indoor environment under human instructions. Previous works ignore the distribution of sample difficulty and we argue that this potentially degrade their agent performance. To tackle this issue, we propose a novel curriculum-based training paradigm for VLN tasks that can balance human prior knowledge and agent learning progress about training samples. We develop the principle of curriculum design and re-arrange the benchmark Room-to-Room (R2R) dataset to make it suitable for curriculum training. Experiments show that our method is model-agnostic and can significantly improve the performance, the generalizability, and the training efficiency of current state-of-the-art navigation agents without increasing model complexity.

## 1  Introduction

Vision-and-Language (VLN) navigation task proposed recently by (Anderson et al., 2018) is a step towards building smart robots. It requires the agent to perceive the environment, understand human language instructions and finally unify the multi-modal information to make actions. Many state-of-the-art methods have been proposed. Some of them focus on the alignment between visual and textual inputs by improving model structure (Ma et al., 2019) or proposing novel auxiliary losses (Zhu et al., 2020) whereas others put their attention on the data augmentation (Fried et al., 2018; Tan et al., 2019; Hong et al., 2020; Ku et al., 2020). Large scale pre-training has also been employed to better generalize the downstream VLN tasks (Li et al., 2019; Majumdar et al., 2020; Hao et al., 2020; Huo et al., 2021).

Despite great progress previous works have made, very few of them care about how much the agent learns from the dataset, i.e. is the agent a good student? In computer vision, (Hlynsson et al., 2019) tries to answer this question by measuring the data efficiency — performance as a function of training set size — of deep learning methods. In vision-and-language navigation, (Huang et al., 2019) develops a discriminator that can filter the low quality instruction-path pairs to boost the learning efficiency. In this work, we focus on another aspect: *can a VLN agent be further educated without model structure change and data modification?*

We observe that many works neglect the internal distribution of the sample difficulty. For example, a navigation task within a single room should be considered easier than one that needs to travel two or more rooms. Current training methods simply flush in the data and do not distinguish difficulty levels among the training samples. As shown in Figure 1(a), navigation agents trained by such learning process do not perform well on those "easy" tasks even on the previously seen environments. We

---

*Corresponding author.

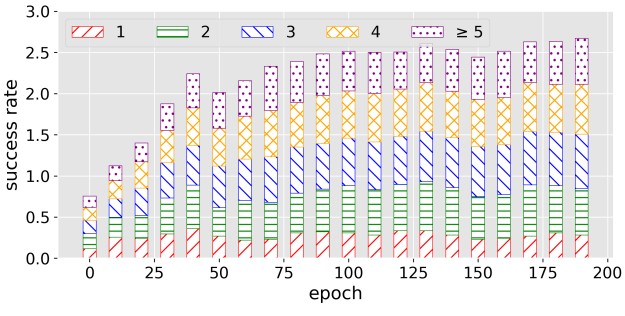 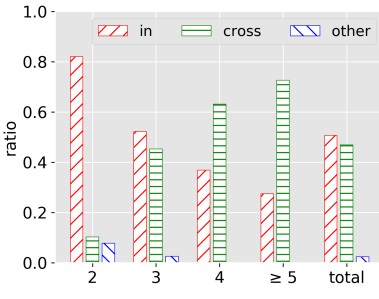

(a) Success rate during training (without curriculum).  (b) Ratio of different errors.

Figure 1: (a) The success rate of Self-Monitoring agent(Ma et al., 2019) on validation seen split of R2R dataset during its training process. Bars with different colors represent the success rate of different sample groups. Numbers in the legend represent the difficulty level of samples, i.e. the navigation task should be completed within how many rooms. (b) The ratio of different types of the **first error** an agent made during navigation. There are three error types where the agent fails to make the right choice. Type "in" represents the correct next viewpoint is within the current room, type "cross" represents the correct next viewpoint is in another room, and type "others" represents that the agent predicts the ground truth trajectory but it fails to stop at the right place.

monitor the first error an agent makes during the navigation and demonstrate the ratio of different types of these errors in Figure 1(b). We find that when the navigation agent fails, about 50% errors are caused by the agent wrongly predicts the next in-room direction. The ratio of this kind of error decreases as the navigation task spans more rooms, but still remains a relatively high level. These phenomenon indicates that the navigation agent is limited by its ability to navigate inside one room and cross two rooms.

The poor performance of agents on those easy cases inspires us to borrow the idea from curriculum learning (Bengio et al., 2009) and propose a curriculum-based training paradigm. The basic idea of curriculum learning is to start small, learn easier aspects of the task and then gradually increase the difficulty level. The definition of "difficulty" is therefore worthwhile. We hypothesise that the difficulty of a navigation task is closely related to the rooms the agent needs to pass by towards the destination. On top of this, we introduce the curriculum design for VLN and re-arrange the benchmark Room-to-Room (R2R) dataset (Anderson et al., 2018) to make it suitable for curriculum learning (Section 3). We incorporate human prior knowledge about training samples into the training process of a navigation agent via self-paced curriculum learning (SPCL) (Jiang et al., 2015). We adapt the traditional SPCL algorithm to efficiently train deep learning models (Section 4). Experiments show that our method can consistently improve both the navigation performance and the training efficiency of navigation agents (Section 5).

In summary, our main contributions are:

- We propose to explicitly incorporate human prior knowledge about training samples into navigation agent training process.
- We design the curriculum for VLN and put forward the training paradigm of a navigation agent by curriculum learning without increasing the complexity of the model.
- We empirically validate that the role of curriculum learning is to smooth loss landscape and hence find a better local optima.

## 2  Related Work

**Vision-and-Language Navigation.**  Vision-and-language Navigation (VLN) is a task where an agent navigates to a goal location in a photo-realistic 3D environment under human instructions. (Anderson et al., 2018) formalized this task, proposed the benchmark Room-to-Room (R2R) dataset and set up an attention-based sequence-to-sequence baseline model. Other related VLN datasets include Touchdown dataset (Chen et al., 2019), which is the first large-scale outdoor VLN dataset,

and CVDN dataset (Thomason et al., 2019) that emphasizes on the robot-human dialogues during navigation.

Embodied navigation tasks suffer from a limited size of training data, which results in several different research focuses. For data augmentation, (Fried et al., 2018) developed a speaker-follower model where the speaker model is usually used as a tool for instruction generation, and (Tan et al., 2019) proposed to apply a effective environmental dropout layer to mimic unseen environments during training. For better generalization, (Wang et al., 2018) introduced a hybrid model that integrated the model-based and model-free reinforcement learning whereas (Wang et al., 2019) proposed a Reinforced Cross-Modal (RCM) agent and a self-supervised imitation learning method to explore the unseen environment. Besides, (Li et al., 2019; Huo et al., 2021; Hao et al., 2020) applied pre-training skills towards better instruction understanding and generalization.

Another resolution to the paucity of training data is rather straightforward: generating more training data and annotations. For example, (Hong et al., 2020) split the instructions in R2R dataset into sub-instructions and annotated the corresponding sub-paths. (Ku et al., 2020) proposed Room Across Room (RxR) dataset, a multilingual VLN dataset with dense spatiotemporal grounding. Furthermore, like (Wang et al., 2020) one can use multitask learning framework to grasp common knowledge from other homologous datasets so as to better the agent's performance on the current task.

These works are worthwhile but we observe that most of them ignore how much the model can learn from the dataset. In this paper, we focus to improve training methods to better exploit data. A better training paradigm can benefit both previous and future works on VLN tasks.

**Curriculum Learning.** Curriculum Learning (Bengio et al., 2009) is a learning paradigm that mimic the learning principle underlying the cognitive process of humans and animals, where a model is learned by gradually including from easy to complex samples in training. (Jiang et al., 2015) bridged the gap between curriculum learning and self-paced learning by proposing a novel self-paced curriculum learning method. (Graves et al., 2017) proposed an automated curriculum learning method that can automatically select the curriculum syllabus. Later, (Matiisen et al., 2017) introduced a teacher-student framework for automatic curriculum learning, where the student tries to learn a complex task and the teacher automatically chooses the task for student to train on.

Curriculum learning has been applied to several natural language processing tasks, such as question answering (Sachan and Xing, 2016; Liu et al., 2018), machine translation (Platanios et al., 2019) and so on. An empirical work done by (Hacohen and Weinshall, 2019) conclude that curriculum learning can effectively modify the optimization landscape and under mild conditions it does not change the corresponding global minimum of the optimization function. These works encourage us to apply the curriculum learning into the VLN tasks as it does not need any modification towards the navigation agent and is capable of improving the agent performance. To the best of our knowledge, this is the first work that introduces curriculum learning purely as a training paradigm for VLN tasks.

Babywalk (Zhu et al., 2020) is another work that applies the curriculum learning idea. It aims to enhance the transfer ability of navigation agents. Therefore, Babywalk uses dynamic programming to decompose the long instructions into shorter ones and then applies these sub-instructions for curriculum-based reinforcement learning. It does not pay much attention to how an agent performs within a dataset. Babywalk is a carefully designed VLN agent with complex model structure whereas our method is a easy, extendable model-agnostic training method. Our method will change neither the difficulty of training data nor the model complexity.

## 3 Curriculum Design for VLN

We observe that different samples in the dataset have different navigation difficulties. Our intuition is that, for human beings it is easy to find an object or a place within a small range. After exploration, it is natural for us to exploit the knowledge about the environment and complete the harder cross-room tasks. Hence, we hypothese that the number of rooms a path could cover (namely room length, $R(p)$) dominates the difficulty level of a navigation task. We therefore propose to re-arrange the R2R dataset based on $R(p)$. Two examples with various $R(p)$ are shown in Figure 2.

We firstly investigate the distributions of rooms that can be covered by a R2R path. Approximately 17% of the paths only pass through less than two rooms whereas about 15% of the paths span 5 rooms

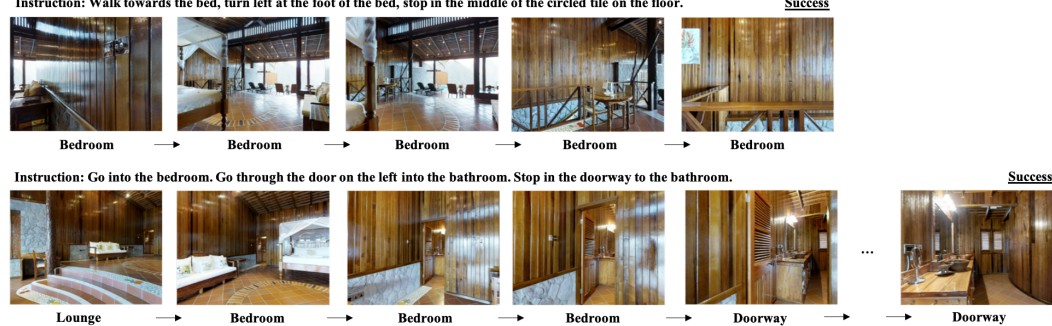

Figure 2: Navigation samples in validation seen split with different room length. Self-Monitoring agent (Ma et al., 2019) succeed to complete the in-room navigation task with trajectory length 6.03 m (above) and the cross-room navigation task with trajectory length 10.17m (below).

Table 1: Basic statistics of CLR2R dataset. Start coverage is defined as the ratio of rooms where the ground truth paths start to all room in the train scans. Room coverage is the ratio of rooms covered by the ground truth paths to all room in the train scans.

| Train Set | Paths | Instructions | Average Trajectory Length (m) | Average Instruction Length (words) | Start Coverage (%) | Room Coverage (%) |
|---|---|---|---|---|---|---|
| R2R | 4675 | 14039 | 9.91 | 29.41 | 89.5 | 96.1 |
| CLR2R | | | | | | |
| - Round 1 | 345 | 1037 | 8.86 | 24.31 | 12.0 | 12.8 |
| - Round 2 | 471 | 1415 | 8.25 | 24.78 | 20.2 | 38.3 |
| - Round 3 | 1632 | 4897 | 9.19 | 28.08 | 59.5 | 79.1 |
| - Round 4 | 1530 | 4593 | 10.42 | 31.16 | 59.5 | 83.0 |
| - Round 5 | 697 | 2097 | 12.12 | 34.33 | 33.5 | 62.2 |

or more. We then propose to split the train set of R2R dataset into 5 mutually exclusive subsets and each subset contains paths whose $R(p)$ are constrained. We believe that the learning from simple to difficult is very similar to play arcade games, so the subsets are named round 1 to 5 according to its difficulty. To be specific, round $i$ ($i = 1, 2, 3, 4$) subset contains samples whose ground truth path covers $i$ rooms. Round 5 subset contains the left samples.

The basic statistics of the re-arranged R2R dataset is summarized in Table 1. As the construction of such dataset incorporate human prior knowledge and such dataset will be futher used for curriculum learning, we call it Room-to-Room for Curriculum Learning (CLR2R) dataset.

It should be noted that the definition of "difficulty" is open and task-specific, we throw a possibility here and encourage other attempts. Similar standards are easy to find in other datasets, for example, the number of dialogue rounds and the richness of dialogue phenomenon can both be taken as difficulty indicators on CVDN dataset(Thomason et al., 2019). Finally, we leave validation and test set of R2R dataset unchanged so as to have comparable experimental results.

## 4 Methods

There are multiple settings of curriculum learning. When a series of tasks is presented, we can apply automated curriculum learning (Graves et al., 2017) where agents are trained by tasks adapted to their capacities. While for single task setting, self-paced curriculum learning (Jiang et al., 2015) is more appropriate if the task dataset has samples with various difficulty levels. Our observations better fit the latter mode, hence we adopt self-paced curriculum learning as the training method.

### 4.1 General Setup

Self-paced curriculum learning (SPCL) is an "instructor-student collaborative" learning method as it considers both prior knowledge known before training and information learning during training in a

Table 2: Different forms of self-paced functions and the corresponding closed form solutions towards Equation 2 when the curriculum region is $[0,1]^n$. $\mathbb{1}_{(\cdot)}$ is the indicator function.

| | Function Form | Closed Form Optimal Solution |
|---|---|---|
| **Binary Scheme** | $g(\boldsymbol{w}; \lambda) = -\lambda \|\boldsymbol{w}\|_1$ | $w_i^* = \mathbb{1}_{(\ell_i \leq \lambda)} \cdot \ell_i$ |
| **Linear Scheme** | $g(\boldsymbol{w}; \lambda) = \frac{1}{2}\lambda \sum_{i=1}^{n} \left(w_i^2 - 2w_i\right)$ | $w_i^* = \mathbb{1}_{(\ell_i \leq \lambda)} \cdot \left(1 - \frac{\ell_i}{\lambda}\right)$ |

unified framework. To be specific, the objective loss function of SPCL is defined as

$$\min_{\boldsymbol{w} \in [0,1]^n, \boldsymbol{\theta} \in \Theta} \mathbb{E}(\boldsymbol{w}, \boldsymbol{\theta}; \lambda, \Psi) \quad = \sum_{i=1}^{n} w_i L\left(y_i, f\left(\boldsymbol{x}_i, \boldsymbol{\theta}\right)\right) + g(\boldsymbol{w}; \lambda) \tag{1}$$
$$\text{s.t.} \qquad \boldsymbol{w} \in \Phi$$

where $f\left(\boldsymbol{x}_i, \boldsymbol{\theta}\right)$ denotes the navigation agent, $\boldsymbol{w} = [w_1, \cdots, w_n]^T \in [0,1]^n$ is the weight variable reflecting the sample importance. $g$ is called the self-paced function which controls the learning scheme, and $\lambda$ is a hyper-parameter that limits the learning pace. $\Phi$ is a feasible region that encodes the information of a predetermined curriculum, defined below.

**Definition 4.1** *For training samples $\boldsymbol{X} = \{\boldsymbol{x}_i\}_{i=1}^n$, given a curriculum $\gamma$ defined on it, the feasible region, defined by,*
$$\Phi = \left\{\boldsymbol{w} \mid \boldsymbol{a}^T \boldsymbol{w} \leq c\right\}$$
*is a curriculum region of $\gamma$ if it holds: 1) $\Phi \wedge \boldsymbol{w} \in [0,1]^n$ is nonempty; 2) $a_i < a_j$ for all $\gamma\left(\boldsymbol{x}_i\right) < \gamma\left(\boldsymbol{x}_j\right); a_i = a_j$ for all $\gamma\left(\boldsymbol{x}_i\right) = \gamma\left(\boldsymbol{x}_j\right)$.*

where vector $\boldsymbol{a} \in \mathbb{R}^n$ that parameterizes a linear space is a function of curriculum $\gamma$. It is not hard to see that, prior human knowledge influence the feasible region $\Phi$ which constrains the weight parameter $\boldsymbol{w}$. Hence, the update of $\boldsymbol{w}$ is subject to the learning progress $L$, self-paced function $g$ and feasible region $\Phi$. Based on our curriculum design, samples within each round should be given the same curriculum rank. Therefore, 5 scalars is enough to define a good vector $\boldsymbol{a}$.

Compared with self-paced learning (SPL) (Kumar et al., 2010), SPCL is more general by introducing self-paced function $g$ as a flexible regularization term. Self-paced functions are defined to be convex, which ensures that we can find good solutions of $\boldsymbol{w}$ within the linear curriculum region. Following (Jiang et al., 2014)'s work, (Jiang et al., 2015) further discussed several examples of self-paced functions. In this paper, we focus on the simplest two: (1) Binary scheme and (2) Linear scheme. As summarized in Table 2, the convex optimization problem

$$\boldsymbol{w}^* = \text{argmin}_{\boldsymbol{w} \in \Phi} \sum w_i \ell_i + g(\boldsymbol{w}; \lambda) \tag{2}$$

have the closed form solution if $\Phi = [0,1]^n$. Therefore, to solve the linear constrained convex optimization problem, we can simply apply a projection gradient descent method to obtain the optimal weight $\boldsymbol{w}^*$.

## 4.2 Algorithm

(Jiang et al., 2015) proposed an alternative convex search (ACS) (M. Bazaraa and Shetty, 1993) algorithm to solve Equation 1. The main problem of the original algorithm is at step 4, where the optimal model parameter $\boldsymbol{\theta}^*$ is learned with most recent weight vector $\boldsymbol{w}^*$ fixed. In the training of navigation agents, it is impossible to compute the exact optimum of $\mathbb{E}(\boldsymbol{w}^*, \boldsymbol{\theta}, \lambda, \Psi)$ due to both the time consumption and the lack of global convergence guarantees. We propose not to compute the exact minimum but replace step 4 with several gradient descent update steps. Doing so benefits the algorithm as the training is much faster then before.

According to (Gorski et al., 2007), the original SPCL algorithm proposed by (Jiang et al., 2015) is guaranteed to converge globally as the objective function is monotonically decreasing and bounded below. Our algorithm does not have the monotonically decreasing property as the stochastic gradient descent (SGD) update can not assure the continuous decrease of the function value. Alternatively,

our algorithm can be viewed as a naive version of mini-batch randomized block coordinate descent (MRBCD) method. (Zhao et al., 2014) claimed that MRBCD using semi-stochastic optimization scheme can attain linear rates of convergence, while the convergence of naive MRBCD is still left to be discussed. Further experiments show that our algorithm is empirically converged, which encourages the theoretical analysis in this direction.

Algorithm 1 takes the input of a predetermined curriculum $\gamma$, an instantiated self-paced function $g$, a stepsize parameter $\mu$ and an update interval $T$; it outputs an optimal model parameter $\boldsymbol{\theta}$.

---

**Algorithm 1** Self-paced Curriculum Learning

---

**Input:** Input dataset $\mathcal{D}$, predetermined curriculum $\gamma$, self-paced function $g$, stepsize $\mu$ and update interval $T$.
**Output:** Model paramater $\boldsymbol{\theta}$.
1: Derive the curriculum region $\Phi$ from $\gamma$.
2: Initialize $\boldsymbol{w}^*$ and $\lambda$ in the curriculum region.
3: **while** not converged **do**
4:     **for** $t = 0, 1, \cdots, T-1$ **do**
5:         Update $\boldsymbol{\theta}_{t+1} = \text{SGD}(\mathbb{E}(\boldsymbol{w}^*, \boldsymbol{\theta}_t; \lambda, \Phi))$
6:     **end for**
7:     Record $\boldsymbol{\theta}^* = \boldsymbol{\theta}_T$
8:     Update $\boldsymbol{w}^* = \arg\min_w \mathbb{E}(\boldsymbol{w}, \boldsymbol{\theta}^*; \lambda, \Phi)$
9:     **If** $\lambda$ is small, **then** increase $\lambda$ by stepsize $\mu$.
10: **end while**
11: **return** $\boldsymbol{\theta}^*$.

---

## 5 Experiment Results and Analysis

### 5.1 Experiment Setup

**Navigation agents.** We experiment with three state-of-the-art VLN agents using different training paradigms and compare their performance on the original R2R validation set. We choose several state-of-the-art navigation agents including the Speaker-Follower agent (Fried et al., 2018) which applies the panoramic action space, the Self-Monitoring agent (Ma et al., 2019) which enforces cross-model alignment and EnvDrop with Back Translation (Tan et al., 2019) that applies reinforcement learning. We reproduce these agents in a unified code framework and replicate the experimental results presented in original papers. We do not use any data augmentation tricks and all three agents are trained within the R2R dataset range.

**Implementation details.** We experiment with three training paradigms. They include the traditional machine learning (ML) strategy that training by uniformly sample mini-batches from R2R dataset, a naive curriculum learning (NCL) strategy that the agent is firstly trained on CLR2R round 1 split, then round 1~2 splits, and gradually trained on the whole CLR2R train set, and previously introduced self-paced curriculum learning (SPCL) strategy. As discussed in Section 4, we set $a_i = i$ for samples in CLR2R round $i$ split. The constant $c$ is chosen within range $[0.95 * \|\boldsymbol{a}\|_1, \|\boldsymbol{a}\|_1]$. For self-paced functions, we choose the binary and linear scheme. For EnvDrop agent (Tan et al., 2019) which is trained by a mixed loss of imitation learning and reinforcement learning, we only use the ground truth trajectory-based loss to update the weight variable. More information is available in supplementary materials.

**Evaluation metrics.** We follow the standard metrics that (Fried et al., 2018) employed for evaluating the agent's performance, including average Navigation Error (NE) which measures the distance between the end location and the target location, Success Rate (SR) which is the percentage of predicted end location within 3m of the target location, and Oracle Success Rate (OSR) which is the percentage of trajectory the shortest distance between which and the target location is within 3m. We also adopt success weighted by path length (SPL) as recommended by (Anderson et al., 2018). Furthermore, we consider the coverage weighted by length score (CLS) (Jain et al., 2019) which measures the overall correspondence between the predicted and ground truth trajectories. Actually,

Table 3: Comparison results on validation unseen and unseen split using different training paradigm. Bolded agent name indicates that it is trained with traditional machine learning method. The plus sign (+) represents the specific curriculum learning method used to train this agent. Evaluation metrics are higher the better except for the navigation error (NE).

| Model | Validation Seen | | | | | Validation Unseen | | | | |
|---|---|---|---|---|---|---|---|---|---|---|
| | NE↓ (m) | SR (%) | OSR (%) | SPL (%) | CLS | NE↓ (m) | SR (%) | OSR (%) | SPL (%) | CLS |
| **Follower** | 4.85 | 52.3 | 65.3 | 44.3 | 58.2 | 7.12 | 28.6 | 40.9 | 20.3 | 35.0 |
| + Naïve CL | 5.03 | 48.6 | 62.0 | 40.4 | 55.9 | 7.13 | 31.1 | 42.8 | 21.2 | 34.3 |
| + Self-Paced CL | **4.23** | **58.7** | **69.2** | **51.1** | **63.3** | **6.69** | **32.2** | **44.2** | **24.5** | **38.9** |
| **Self-Monitoring** | 4.27 | 58.4 | 67.0 | 51.9 | 64.1 | 6.42 | 38.4 | 48.1 | 28.3 | 41.5 |
| + Naïve CL | **4.08** | **61.0** | **69.8** | **54.6** | 64.9 | 6.30 | 40.0 | 51.7 | 28.6 | 41.1 |
| + Self-Paced CL | 4.19 | 58.8 | 68.2 | 53.3 | **65.4** | **5.98** | **41.0** | **52.4** | **30.8** | **43.9** |
| **EnvDrop** | 4.55 | 57.7 | **65.6** | 54.4 | 67.2 | 5.92 | 45.7 | 54.2 | 41.8 | 57.0 |
| + Naïve CL | 4.49 | 57.8 | 63.1 | **54.8** | 67.2 | 5.93 | 44.3 | 50.5 | 41.3 | 57.6 |
| + Self-Paced CL | **4.42** | **58.1** | **65.6** | **54.8** | **67.4** | **5.48** | **47.6** | **54.3** | **44.1** | **59.1** |

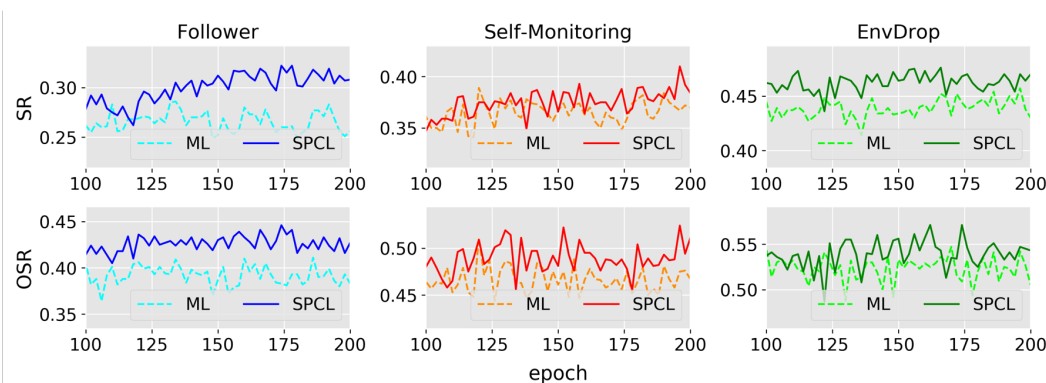

Figure 3: Success rate (SR) and oracle success rate (OSR) of navigation agents trained by machine learning (ML) and self-paced curriculum learning (SPCL) on validation unseen split.

nDTW and SDTW proposed by (Magalhães et al., 2019) are both useful metrics that capture path fidelity, hence we supplement a full-metric version of main results in supplementary materials.

## 5.2 Overall Performance

We first compare the performance of the different training paradigms on the CLR2R(R2R) validation set. Table 3 shows overall results. Experiments show that naïve curriculum learning is surprisingly effective. It improves the navigation outcomes of Follower and Self-Monitoring agent on validation unseen split. In contrast to machine learning where the whole training set is available, naïve curriculum learning exposes the training samples sequentially from easy to hard. Hence, follower agent trained by NCL is underperformed on the validation seen split due to its limited learning ability. For self-paced curriculum learning, we initialize the algorithm to pay more attention on round 1 and 2 splits, then allow it actively assigns higher weight for examples which the agent learns better. During training, the weight for each sample can finally converge nearly to one. Experiments show that agents trained by SPCL consistently achieve good performance on both validation seen and unseen split.

Furthermore, we compare the learning speed of agents trained by different training paradigms. As shown in Figure 3, there is obviously a gap between agents trained by machine learning and self-paced curriculum learning. Agents trained by curriculum learning can learn fast and achieve better performance after the same number of iterations. This indicates that self-paced curriculum learning can not only improve the performance, but also optimize the training efficiency of the agents.

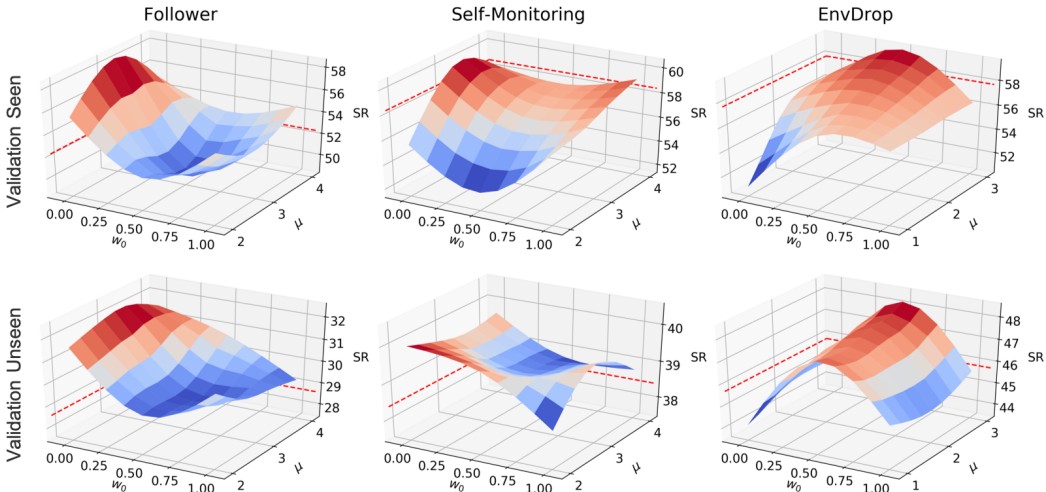

Figure 4: Success rate (SR) on CLR2R(or R2R equivalently) validation set with different SPCL hyper-parameter settings. The self-paced function is fixed as the linear scheme. Red dashed line represents the result from agents trained by machine learning. The figure is drawn by cubic interpolation.

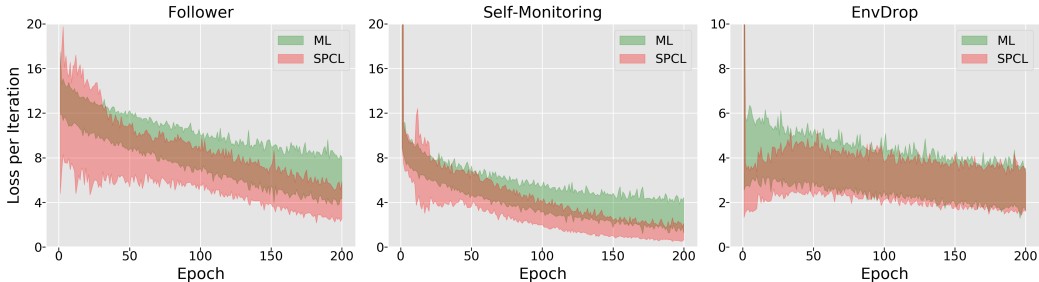

Figure 5: Loss landscape of normal machine learning (ML) and self-paced curriculum learning (SPCL) trained agents. At a particular training epoch, we record the maximum and minimum iteration loss, i.e. an average loss of batch samples. We plot the distances between the two as the shadowed area between lines.

## 5.3 Further Analysis

We also investigate the hyperparameter robustness of self-paced curriculum learning. Moreover, we present the transfer learning outcomes of curriculum learning method on CLR2R and RxR dataset.

**Hyperparameter robustness.** To understand how the weight initialization and stepsize choice influence the training paradigm, we grid search these two hyperparameters and the result is shown in Figure 4. Following the core idea of curriculum learning by starting small, we enforce the initial weight for round 1 and 2 samples as one. Then $w_0$ represents the initial weight for samples in round 3~5 splits. As shown in Figure 4, SPCL favors a smaller $w_0$. This validates the principle of curriculum learning. Besides, SPCL is rather robust to weight initialization and the choice of stepsize as in most cases, the results in validation unseen split are better than the machine learning baseline. This also indicates a better generalizability of agents trained by SPCL. For validation seen split, the performance is usually no worse than the baseline. The overall outcomes prove that curriculum learning can improve the data exploitation efficiency and enhance the generalization ability.

**Loss landscape.** Following (Santurkar et al., 2018), we investigate the loss landscape during training by computing the distance between maximum and minimum loss. As shown in Figure 5, SPCL narrows the loss gap for Follower and Self-Monitoring agent. For EnvDrop agent, SPCL differs little from ML. We attribute this to the mixed learning strategy of EnvDrop, whose loss is a weighted sum of imitation learning(0.2) and reinforcement learning(0.8). Hence, the power of SPCL is not as

Table 4: Basic statistics of R2R and RxR dataset.

| Dataset | Average Path Length (m) | Average Edges | Average Instruction Length (words) |
|---|---|---|---|
| R2R (Anderson et al., 2018) | 9.4 | 5 | 29 |
| RxR (Ku et al., 2020) | 14.9 | 8 | 78 |

Table 5: Transfer learning results of Self-Monitoring agent on validation unseen split. Character F and S in Line (4) represents "first" and "second" respectively. RxR is short for RxR-en subset of RxR datset in this table.

| | Train Set | | Validation Unseen | | | | | | | | | |
|---|---|---|---|---|---|---|---|---|---|---|---|---|
| | | | NE(m) | | SR(%) | | OSR(%) | | SPL(%) | | nDTW | |
| | R2R | RxR | R2R | RxR | R2R | RxR | R2R | RxR | R2R | RxR | R2R | RxR |
| (1) | ✓ | | **6.42** | 10.39 | **38.4** | 16.0 | **48.1** | 29.9 | **28.3** | 9.1 | **43.7** | **28.0** |
| (2) | | ✓ | 7.72 | **10.12** | 16.7 | **18.7** | 30.4 | **32.8** | 5.9 | **6.9** | 18.6 | 18.2 |
| (3) | ✓ | ✓ | 6.66 | 10.17 | 32.7 | 20.5 | **45.8** | 33.7 | 22.4 | 10.9 | 36.4 | 23.9 |
| (4) | F | S | 6.96 | 9.91 | 26.4 | **22.1** | 38.6 | **34.8** | 18.0 | **14.3** | 36.8 | **29.8** |
| (5) | Naïve CL | | **6.46** | **9.90** | **36.3** | 21.6 | 44.4 | 33.4 | **27.4** | 13.5 | **43.0** | 29.3 |

significant as in other agents. In general, our experimental results correspond to the theoretical result (Hacohen and Weinshall, 2019) that curriculum learning can effectively smooth the optimization landscape.

**Transfer learning.** To explore whether curriculum learning helps the agent to obtain better transferring ability, we conduct the following experiment as summarized in Table 5. As mentioned in Section 2, RxR dataset (Ku et al., 2020) is a large-scale multilingual VLN dataset built upon Matterport3D simulator (Chang et al., 2017). We select the RxR dataset as the target dataset of transfer learning since it shares the same environment with R2R dataset but is much harder in many aspects (see Table 4). As RxR dataset is multilingual, we employ its english subset, RxR-en, to avoid the language bias. For fairly comparison, the number of training iterations is fixed at 80,000 for line (3)~(5) in table. Specifically, for line (3) the agent is trained on a integrated R2R+RxR-en dataset for 80,000 iterations; for line (4), the agent is firstly trained on R2R trainset for 40,000 iterations and then transferred to RxR-en dataset for another 40,000 iterations; for line (5), the agent is firstly trained on R2R trainset and then trained on the integrated R2R+RxR-en dataset for another 40,000 iterations.

Line (1) and (2) indicates that transfer learning (R2R→RxR and vice-versa) performance is much weaker than the in-domain results. For line (3), the agent trained on an integrated dataset performs better on RxR-en validation split, but its performance on R2R validation split it not as good as line (1). The difficulty gap between samples from two data source confuses the agent. Line (3) and (4) indicates that starting from simple improves the agents performance on RxR dataset. Compared with line (4), curriculum learning in line (5) helps the agent to preserve its navigation ability on R2R dataset. Actually, line (4) can also be taken as a curriculum learning experiment whose target distribution is RxR dataset instead of R2R and RxR dataset.

This exploration experiment suggests that to transfer from an easier dataset towards a harder one, curriculum learning paradigm helps the agent to improve the performance on the harder dataset without degrading the performance on the easier dataset.

**Combination with Pre-training.** Pre-training methods are proved strong in many areas. We believe that navigation agents trained by our proposed curriculum training paradigm can also benefit from pre-training skills. That is, curriculum learning does not conflict with pre-training. To validate this, we combine agents trained by curriculum based methods with VLN-BERT (Majumdar et al., 2020), a visiolinguistic transformer-based model. VLN-BERT is a scoring function that evaluates the compatibility between path-instruction pairs, hence it can be easily applied to assist our navigation agents. Since VLN-BERT has to be used under beam search settings , we implement beam search only here. In (Majumdar et al., 2020), authors use a beam size as 30 to include as much path candidates as possible. While in this experiment, we aim to evaluate the usefulnees of combining curriculum and pre-training. Therefore, we restrict the beam size as 5 and purely use VLN-BERT model to scoring

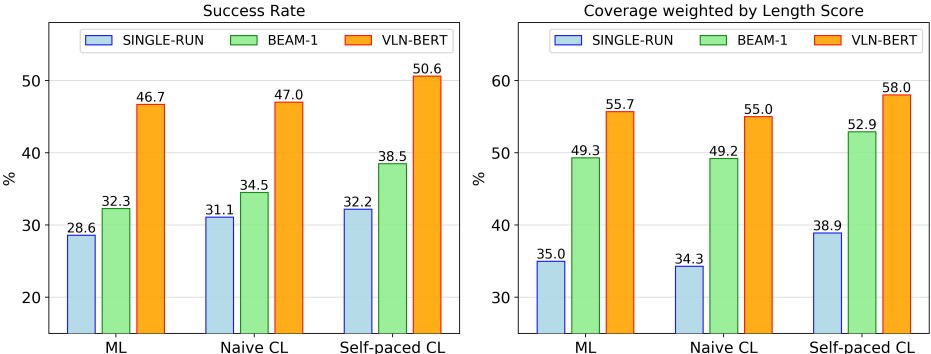

Figure 6: Success rate and coverage weighted by length score of Follower model trained by normal machine learning (ML), naive curriculum learning (Naive CL), self-paced curriculum learning (Self-paced CL) on validation unseen split under three different test modes. SINGLE-RUN means always using argmax in action selection. BEAM-1 means beam search with beam size as 1. VLN-BERT means the path-instruction pairs generated by beam search with size as 5 are scored purely by VLN-BERT model.

and selecting the path-instruction pairs. We experiment with Follower agent under three different test modes, i.e. single-run, beam search with size 1, and vln-bert. Results are shown in Figure 6. Beam search and VLN-BERT scorer both can improve the agent performance. Navigation agents trained by curriculum-based methods obtain more improvements.

## 6    Conclusion and Future Works

We propose to apply curriculum learning as a useful training paradigm of VLN agents. We adapt the traditional self-paced curriculum learning and design the first curriculum for VLN task based on R2R dataset. Experiments illustrate that our method is model-agnostic and can improve both the performance and learning efficiency of the navigation agents. We verify the power of curriculum learning comes from its ability to smooth the loss landscape. Our further exploration indicates that curriculum learning is suitable for transfer learning and combination with pre-training methods.

In the future, we would like to explore the VLN task in two directions. Along the path of this paper, we would like to explore curriculum learning for vision-language navigation in a different scenario of city traveling where the agent needs to perform multi-scale navigation, i.e. city-level and building-level. Alternatively, inspired by the ampleness of VLN tasks and datasets, i.e. R2R (Anderson et al., 2018), CVDN (Thomason et al., 2019), HANNA (Nguyen and Daumé, 2019), TOUCHDOWN (Chen et al., 2019), REVERIE (Qi et al., 2020) and so on, it is possible to apply meta-learning on top of these datasets and get an adaptable navigation agent.

## Acknowledgements

This work is partially supported by Natural Science Foundation of China (No.71991471, No.6217020551), Science and Technology Commission of Shanghai Municipality Grant (No.20dz1200600, 21QA1400600) and Zhejiang Lab (No. 2019KD0AD01).

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
