# A   Appendix

## A.1   Training Details

We train all agents on a single NVIDIA GeForce RTX2080 GPU, using the same model hyperparameters as the officially released codes, except the language encoder of Follower agent (Fried et al., 2018) where we enforce a bidirectional LSTM module and we do not use GloVe embeddings as initialization. For fairly comparison, basic training hyperparameters are fixed, i.e. the maximum training epoch is 200. We sample mini-batches with size 64 for 200 iterations per epoch. The learning rate is a constant and is fixed at $1e^{-4}$.

With regard to self-paced curriculum learning, we choose linear scheme with $w_0 = 0.0, \mu = 3.0$ for Follower agent, binary scheme with $w_0 = 1.0, \mu = 3.0$ for Self-Monitoring agent, and linear scheme with $w_0 = 0.5, \mu = 2.0$ for EnvDrop agent. Value $\lambda$ is initialized as 2 for all agents. It is updated by $\mu$ if it is smaller than currently maximum item loss and is updated by half of $\mu$ otherwise.

Table 6: Comparison results on validation set and test set using different training paradigm. All evaluation metrics are reported.

| Model | Validation Seen | | | | | | |
|---|---|---|---|---|---|---|---|
| | NE $\downarrow$ | SR $\uparrow$ | OSR $\uparrow$ | SPL $\uparrow$ | nDTW $\uparrow$ | SDTW $\uparrow$ | CLS $\uparrow$ |
| **Follower** | 4.85 | 52.3 | 65.3 | 44.3 | 59.4 | 44.1 | 58.2 |
| + Naïve CL | 5.03 | 48.6 | 62.0 | 40.4 | 57.4 | 40.5 | 55.9 |
| + Self-Paced CL | **4.23** | **58.7** | **69.2** | **51.1** | **64.5** | **50.5** | **63.3** |
| **Self – Monitoring** | 4.27 | 58.4 | 67.0 | 51.9 | 65.4 | 51.5 | 64.1 |
| + Naïve CL | **4.08** | **61.0** | **69.8** | **54.6** | 66.2 | **53.9** | 64.9 |
| + Self-Paced CL | 4.19 | 58.8 | 68.2 | 53.3 | **66.3** | 52.2 | **65.4** |
| **EnvDrop** | 4.55 | 57.7 | **65.6** | 54.4 | 67.2 | 51.4 | 67.2 |
| + Naïve CL | 4.49 | 57.8 | 63.1 | **54.8** | 67.5 | 51.5 | 67.2 |
| + Self-Paced CL | **4.42** | **58.1** | **65.6** | **54.8** | **67.7** | **51.6** | **67.4** |

| Model | Validation Unseen | | | | | | |
|---|---|---|---|---|---|---|---|
| | NE $\downarrow$ | SR $\uparrow$ | OSR $\uparrow$ | SPL $\uparrow$ | nDTW $\uparrow$ | SDTW $\uparrow$ | CLS $\uparrow$ |
| **Follower** | 7.12 | 28.6 | 40.9 | 20.3 | 37.0 | 20.8 | 35.0 |
| + Naïve CL | 7.13 | 31.1 | 42.8 | 21.2 | 36.9 | 22.1 | 34.3 |
| + Self-Paced CL | **6.69** | **32.2** | **44.2** | **24.5** | **40.9** | **24.5** | **38.9** |
| **Self – Monitoring** | 6.42 | 38.4 | 48.1 | 28.3 | 43.7 | 28.8 | 41.5 |
| + Naïve CL | 6.30 | 40.0 | 51.7 | 28.6 | 43.4 | 29.4 | 41.1 |
| + Self-Paced CL | **5.98** | **41.0** | **52.4** | **30.8** | **46.0** | **31.0** | **43.9** |
| **EnvDrop** | 5.92 | 45.7 | 54.2 | 41.8 | 56.7 | 39.3 | 57.0 |
| + Naïve CL | 5.93 | 44.3 | 50.5 | 41.3 | 57.3 | 38.3 | 57.6 |
| + Self-Paced CL | **5.48** | **47.6** | **54.3** | **44.1** | **59.3** | **41.2** | **59.1** |

| Model | Test | | | |
|---|---|---|---|---|
| | NE $\downarrow$ | SR $\uparrow$ | OSR $\uparrow$ | SPL $\uparrow$ |
| **Follower** | 7.05 | 29.0 | 41.3 | 20.7 |
| + Naïve CL | 7.16 | 28.3 | 40.9 | 19.6 |
| + Self-Paced CL | **6.95** | **30.9** | **42.3** | **24.3** |
| **Self – Monitoring** | 6.29 | 40.5 | 50.2 | **30.9** |
| + Naïve CL | **6.29** | **40.8** | **53.0** | 30.6 |
| + Self-Paced CL | 6.29 | 39.3 | 49.9 | 30.8 |
| **EnvDrop** | 5.71 | 46.5 | **54.2** | 43.5 |
| + Naïve CL | 5.90 | 44.8 | 50.0 | 42.5 |
| + Self-Paced CL | **5.41** | **48.4** | 53.9 | **45.5** |

## A.2   Full Metric Results

As coverage weighted by length score (CLS) (Jain et al., 2019) which measures the overall correspondence between the predicted and ground truth trajectories, other metrics like normalized dynamic timewarping (nDTW) and success weighted by normalized dynamic time warping (SDTW) can also capture the path fidelity. Therefore, to give readers a general picture, we supplement a full-metric version here on validation set (see Table 6).

Besides, we also supplement the results on test set. However, due to the limit of EVAL platform, results on test set only have 4 evaluation metrics and hence is not presented in paper.

## A.3 Exploration Experiments

**Randomness Check**   Except for the main results reported in paper, we repeat the experiments for 4 more times and every time we use the same training settings except the random seed. Table 7 summarizes our results on validation unseen split. The agent's performance is consistent with paper.

Table 7: Mean and standard error on validation unseen split from repeating experiments. Standard errors are in brackets.

| Model | Validation Unseen | | | | |
|---|---|---|---|---|---|
| | NE ↓ | SR ↑ | OSR ↑ | SPL ↑ | nDTW ↑ |
| **Follower** | 6.98 (0.14) | 29.71 (1.68) | 40.75 (1.45) | 20.66 (1.46) | 37.10 (1.15) |
| + Naïve CL | 7.03 (0.07) | 29.95 (0.84) | 42.92 (1.34) | 20.01 (0.99) | 36.52 (0.81) |
| + Self-Paced CL | **6.75** (0.11) | **32.08** (0.77) | **40.39** (1.16) | **23.83** (0.86) | **40.39** (1.10) |
| **Self − Monitoring** | 6.35 (0.07) | 38.22 (1.32) | 48.65 (2.01) | 29.15 (1.73) | 44.62 (1.42) |
| + Naïve CL | 6.35 (0.06) | 39.83 (0.43) | **50.44** (1.60) | 29.77 (1.30) | 44.44 (1.53) |
| + Self-Paced CL | **6.18** (0.14) | **40.22** (1.33) | 50.05 (1.96) | **32.12** (1.16) | **47.62** (1.07) |
| **EnvDrop** | 5.79 (0.09) | 45.44 (0.28) | 54.11 (0.89) | 41.84 (0.30) | 57.69 (0.61) |
| + Naïve CL | 5.91 (0.09) | 44.66 (0.84) | 51.91 (0.97) | 41.46 (0.72) | 57.28 (0.42) |
| + Self-Paced CL | **5.71** (0.15) | **46.11** (0.89) | **54.13** (1.21) | **42.52** (0.98) | **58.18** (0.91) |

**Order Check**   From the results in Table 3, naive curriculum method has negligible gains on EnvDrop model, who uses a mixed learning strategy, when compared with other two models. To further ensure the success on Follower and Self-Monitoring models does not come from side effect of simple ordering of samples, we conduct an experiment where samples are organized by number of rooms in the path and the feeding order are randomly selected from 1 to 5. In Table 8, the success rate of different agents presents a consistent decreasing trend when the input order is randomly selected.

Table 8: Results on three navigation models by randomly selecting the input order. To better compare with the previous results, we also include the performance of agents trained by normal and naive curriculum based methods. Metrics are higher the better except for the navigation error (NE).

| Model | Validation Seen | | | | | Validation Unseen | | | | |
|---|---|---|---|---|---|---|---|---|---|---|
| | NE ↓ | SR | OSR | SPL | nDTW | NE ↓ | SR | OSR | SPL | nDTW |
| **Follower** | 4.85 | 52.3 | 65.3 | 44.3 | 59.4 | 7.12 | 28.6 | 40.9 | 20.3 | 37.0 |
| + Random | 5.27 | 47.6 | 61.8 | 37.2 | 52.2 | 7.00 | 28.1 | 40.8 | 18.8 | 35.9 |
| + Reverse CL | 4.82 | 51.2 | 67.5 | 42.4 | 58.2 | 7.04 | 28.9 | 41.6 | 19.1 | 35.6 |
| + Naïve CL | 5.03 | 48.6 | 62.0 | 40.4 | 57.4 | 7.13 | 31.1 | 42.8 | 21.2 | 36.9 |
| **Self − Monitoring** | 4.27 | 58.4 | 67.0 | 51.9 | 65.4 | 6.42 | 38.4 | 48.1 | 28.3 | 43.7 |
| + Random | 5.08 | 53.5 | 64.1 | 45.6 | 59.5 | 6.79 | 36.4 | 46.4 | 26.3 | 40.3 |
| + Reverse CL | 4.32 | 58.5 | 71.5 | 52.1 | 65.6 | 6.49 | 38.8 | 52.3 | 27.8 | 43.1 |
| + Naïve CL | 4.08 | 61.0 | 69.8 | 54.6 | 66.2 | 6.30 | 40.0 | 51.7 | 28.6 | 43.4 |

Furthermore, we reverse the input order, that is, the feeding order is monotonically decreasing, i.e. the agent is gradually trained on round 5 split, round 5~4 splits, round 5~3 splits and finally the whole CLR2R dataset. For follower agent, results show that the best success rate for seen and unseen split are 51.2% and 28.9% respectively, which is very close to the normally trained agent. For self-monitoring agent, the reverse experiment gives 58.5% and 38.8% success rate respectively and such performance is also close to the normally trained agent (but lower than the CL trained agent). Hence, it is the curriculum training paradigm that contributes to the good performance.

**Extension to FGR2R**   Sub-instruction and sub-paths can be considered simpler navigation tasks and hence they are naturally suitable to be included in a CL framework. We combine both R2R (Anderson et al., 2018) and FGR2R dataset (Hong et al., 2020), then traine an agent using naive CL method on that. Specifically, we split the training samples from R2R and FGR2R datasets into 3 splits using the same room-coverage heuristic. These splits contains instruction-path pairs for 1 room,

2 rooms, and $\geq 3$ rooms respectively. As shown in Table 9, with more information given, curriculum learning paradigm can further improve the agent's performance.

Table 9: Results on Follower model using different training methods. Evaluation metrics are higher the better except for the navigation error (NE).

| | Validation Seen | | | | | Validation Unseen | | | | |
|---|---|---|---|---|---|---|---|---|---|---|
| | NE ↓ | SR | OSR | SPL | nDTW | NE ↓ | SR | OSR | SPL | nDTW |
| Normally w/ R2R | 4.85 | 52.3 | 65.3 | 44.3 | 59.4 | 7.12 | 28.6 | 40.9 | 20.3 | 37.0 |
| Naive CL w/ CLR2R | 5.03 | 48.6 | 62.0 | 40.4 | 57.4 | 7.13 | 31.1 | 42.8 | 21.2 | 36.9 |
| Naive CL w/ FGR2R | **4.41** | **57.5** | **70.1** | **49.8** | **62.6** | **6.79** | **32.4** | **43.8** | **24.0** | **40.3** |