# OpenReview forum: "Curriculum Learning for Vision-and-Language Navigation"
_NeurIPS.cc/2021/Conference — NeurIPS 2021 Poster_

### Official Review · Reviewer_mN5T · 2021-07-15

**Rating:** 5
**Confidence:** 4

**Summary:**

This paper applies self-paced curriculum learning [1] to VLN. It presents two approaches: a naive approach where the agent is trained on data with increasing coverage over the environment, and a learning-based approach that dynamically adapts the weights of learning examples according to the agent's learning progress. On three types of VLN agents, these approaches yield moderate improvement `(~2-3% improvement in success rate on validation unseen environments).

**Limitations And Societal Impact:**

The curriculum is designed based on the path structure only. It would be nice to measure and include the complexity of the language in the instructions in designing the curriculum.

**Main Review:**

**Originality**: the technique is primarily based on self-paced curriculum learning by [1] with minor adaptions to a mini-batch setting. The novelty is mostly in the empirical results and analyses. The paper demonstrates potential benefits (in both success rate and learning efficiency) of enforcing a curriculum in training VLN agents. Another interesting result is that pre-training an agent with the (easier) R2R tasks can help with learning the (more difficult) RxR tasks.

[1] Self-paced curriculum learning. https://www.cs.cmu.edu/~lujiang/camera_ready_papers/AAAI_SPCL_2015.pdf

**Quality**:

The SPCL technique is based on a well-established prior work and thus is solid. However, I have a few concerns about the experimental procedure.

First of all, I notice that only validation results are reported. If the validation sets are used to tune the agents, then these results do not indicate the generalizability of the agents, and thus are not sufficient to demonstrate the effectiveness of the proposed technique. I request the authors to include results on the test sets.

Second, because there are randomness in various parts of the proposed technique (due to the mini-batch setting), the authors should also repeat the experiments with multiple random seeds and report error bars of the results. Results as those in Figure 3 may not hold under a different initialization configuration.

In addition, the messages conveyed by Figure 1 are not very clear to me. In Figure 1a, to illustrate the benefits of training with a curriculum, there needs to be another contrasting Figure showing the same plot but with curriculum learning. Figure 2b shows that the agent "is limited by its ability to navigate inside one room" but I am not why that motivates curriculum learning. It makes more sense to me to plot the distribution of errors over the course of training. If we see that at the beginning, in-room errors dominate (in percentage) then it is reasonable to prioritizing training examples that can fix these errors.

As discussed in the paper, we should expect that the self-paced curriculum increasingly includes more examples over time. It would be nice to verify that (e.g. by visualizing the norm of w over time).

**Clarity**:

The paper is straightforward to read. However, the descriptions of the SPCL leaves out a few important details and intuitions. In line 144 and even in the appendix, the *a* vector is not clearly defined. The update of lambda in algorithm 1 should be more detailed. When is lambda "small"? and what is the formulation for updating the lambda? This sentence seems to be copied and pasted from the SPCL's original paper. In this paper, you need to provide sufficient detail so that other researchers can reproduce the results.

I also suggest the authors give a more detailed descriptions of the agents, specifically about their architectural differences. More importantly, are the architectural differences the only differences? Are the agents trained under the same data in each experiment? For example, the EnvDrop agent is trained with environment dropout in its paper; and the Follower agent is trained with additional back-translated data in its paper. Do you also include those data in training them in this paper?

In figure 1, you may want add to the caption that the results are obtained without curriculum learning.

A minor thing is to put up and down arrows next to the metrics in the result tables so that readers can quickly identify which direction is better.

**Significance**: overall, the gain the validation data is moderate. This paper represents an incremental step towards solving VLN.


**Time Spent Reviewing:**

5 hours

---

> ### Author Response · Authors · 2021-08-09
> **Reply to Reviewer mN5T**
>
> Thanks for all the constructive comments. Here is our reply to your questions.
>
> Q1:  Validation sets are not sufficient to demonstrate the effectiveness of the proposed technique. I request the authors to include results on the test sets.
>
> A1:  Due to the submission limit of EvalAI platform, complete results of testset will be presented in the next version of our paper. For now, we have the results for the Follower model, i.e. the success rate of normally trained and spcl trained agent are 29.0% and 30.9% respectively, the oracle success rate are 41.3% and 42.3% respectively and success rate weighted by path length are 20.7% and 24.3% respectively. The performance gap indicates the effectiveness of CL paradigm.
>
>
>
> Q2:  There are randomness in various parts of the proposed technique (due to the mini-batch setting), the authors should also repeat the experiments with multiple random seeds and report error bars of the results.
>
> A2: Repeating the experiments is time-costing and we haven't finished all. Currently available results is based on the follower model and the mean and std error of success rate on the validation unseen split for the normally trained, naive CL trained and SPCL trained agent are 29.7%(±1.7%), 30.0%(±0.8%) and 32.1%(±0.8%) respectively. The oracle success rate on the validation unseen split for three differently trained agents are 40.8%(±1.5%), 42.9%(±1.3%) and 43.2%(±1.2%) respectively. Note that we repeated the experiment for 5 times and every time used the same training settings except the random seed. Complete results will be available in the next version of our paper.
>
>
>
> Q3:  The messages conveyed by Figure 1 are not very clear to me. In Figure 1a, to illustrate the benefits of training with a curriculum, there needs to be another contrasting Figure showing the same plot but with curriculum learning. Figure 2b shows that the agent "is limited by its ability to navigate inside one room" but I am not why that motivates curriculum learning. It makes more sense to me to plot the distribution of errors over the course of training. If we see that at the beginning, in-room errors dominate (in percentage) then it is reasonable to prioritizing training examples that can fix these errors.
>
> A3:  Figure 1 is to demonstrate a phenomenon that we find during the replication experiment of Self-Monitoring agent, which inspired the idea of useing CL method.  It is reasonable to add another contrasting figure, in the next version. For Figure 1b, we present the ratio of different errors of a trained agent because for an undertrained agent during the course of training, experiments show that it is highly likely to make a mistake within a few steps after the beginning. Hence, at the beginning, there is little chance that other types of errors occur. We think that it is more important to study the reasons why a mature agent makes mistakes.
>
>
>
> Q4:  The descriptions of the SPCL leaves out a few important details and intuitions. In line 144 and even in the appendix, the *a* vector is not clearly defined. The update of lambda in algorithm 1 should be more detailed. When is lambda "small"? and what is the formulation for updating the lambda? This sentence seems to be copied and pasted from the SPCL's original paper. In this paper, you need to provide sufficient detail so that other researchers can reproduce the results.
>
> A4:  Actually, a vector is simply a n-dim vector that parameterizes a linear space. With regard to the update of lambda, in our implementation it is initialized as a small number, i.e. 2 for follower and envdrop and 4 for self-monitor. Lambda will be updated by stepsize miu when it is lower than then the maximum loss for a single sample, otherwise it is updated by half of miu. The value of miu is different for different models, in our implementation it is 2 for monitor and envdrop and 3 for follower. For lambda, the "small" or "large" is decided by the comparison with 25%, 50% and 75% quantile of loss. We say lambda is small if it is lower than the 25% quantile. We will add more details about the model training and tuning in the new version of our paper.
>
>
>
> Q5:  In figure 1, you may want add to the caption that the results are obtained without curriculum learning.
>
> A5:  Yes, we will emphasize on this point.
>
>
>
> Q6:  A minor thing is to put up and down arrows next to the metrics in the result tables so that readers can quickly identify which direction is better.
>
> A6: Thanks for the advice. We will do so in the new version of our paper.

---

> > ### Comment · Reviewer_mN5T · 2021-08-27
> > **Thank you for the response!**
> >
> > Thank you for the response. It clarifies most of my points. I raised the score to 5.

---

### Official Review · Reviewer_uKaH · 2021-07-17

**Rating:** 6
**Confidence:** 5

**Summary:**

The paper proposes a curriculum learning (CL) based approach for the task of Vision-Language Navigation (VLN). The dataset chosen for experimentation is R2R which involves an agent to navigate from room to room to follow the provided instruction. The proposed method creates a curriculum for the learning agent based on the number of rooms a given path covers. The empirical results are reported using two methods -- (1) naive CL where the agent is trained on gradually harder tasks, (2) self paced curriculum learning (SPCL).



**Main Review:**

Strengths:

-- The criteria for creating a curriculum is easy and straightforward for the given dataset.

-- The main empirical results show clear gains due to the SPCL method.

-- The paper is clearly written and the technical contributions are neatly presented. Fig 3,4,5 have good visualizations.

Weaknesses:

-- The contributions of this paper are limited to empirical studies. The concept of CL has been studied in this domain and shown to have value in Zhu et al. The methodology (SPCL) is the same as Jiang et al.

-- From the results in Table 3, there are negligible gains due to CL on EnvDrop method. Since EnvDrop method obtains the best results (out of the 3 studied methods) on R2R dataset, it is unclear if CL is contributing to the success of other two methods or whether the other two methods benefit due to a side effect of using some simple ordering of example visitation. For instance, if an experiment is conducted where the examples are fed using number of rooms in the path as criteria but the feeding order is random instead of monotonically increasing from 1 to 5. Will a random order perform similar to the curriculum learning order?

-- The criteria for creating a curriculum seems highly tailored for the given R2R dataset. Such criteria won't be extendible to other VLN datasets like CVDN or Touchdown.

======= Post author response updates ==========

Increasing my score from 5 to 6. See my response to authors' response for rationale.

**Time Spent Reviewing:**

2

---

> ### Author Response · Authors · 2021-08-09
> **Reply to Reviewer uKaH**
>
> Thanks for all the constructive comments. Here is our reply to your questions.
>
> Q1:  The contributions of this paper are limited to empirical studies. The concept of CL has been studied in this domain and shown to have value in Zhu et al.
>
> A1: Our contribution is to show that we should look into the distribution of training data so as to better utilize them. With regard to Zhu et al, they use dynamic programming to decompose the long instructions into shorter ones and then applies these sub-instructions for curriculum-based reinforcement learning. That is, Zhu et al creates the easier samples to form a curriculum instead of investigating the training data as we have done. Another difference is that, Zhu et al relate the length of instructions to the difficulty of task whereas we think the room in path makes sense. Besides, our method is a model-agnostic and can be easily inserted to the training process of other VLN models.
>
>
>
> Q2:  From the results in Table 3, there are negligible gains due to CL on EnvDrop method. Since EnvDrop method obtains the best results (out of the 3 studied methods) on R2R dataset, it is unclear if CL is contributing to the success of other two methods or whether the other two methods benefit due to a side effect of using some simple ordering of example visitation. For instance, if an experiment is conducted where the examples are fed using number of rooms in the path as criteria but the feeding order is random instead of monotonically increasing from 1 to 5. Will a random order perform similar to the curriculum learning order?
>
> A2: With regard to your concern, we did two experiment. Firstly, to check whether the success of other two methods comes from side effect of simple ordering of samples, we did an experiment where samples are organized by number of rooms in the path and the feeding order are randomly selected from 1~5. Results showed that, for follower model, after training 200 eps the success rate for seen and unseen split are 48.5% and 27.7% respectively. Hence, the performance gap is not caused by the side effect of simple ordering. Secondly, we reverse the input order, that is, the feeding order is monotonically decreasing from 5 to 1 and see what happens. Results show that the best success rate for seen and unseen split are 53.5% and 28.9% respectively, which is very close to the normally trained follower agent. For self-monitoring agent, the reverse experiment gives 58.5% and 38.8% success rate respectively and such performance is also close to the normally trained follower agent (but lower than the CL trained agent). Therefore, we can say that it is CL method that contributes to the good performance. More experimental outcomes will be updated in the paper.
>
>
>
> Q3:  The criteria for creating a curriculum seems highly tailored for the given R2R dataset. Such criteria won't be extendible to other VLN datasets like CVDN or Touchdown.
>
> A3:  A:  As we have mentioned in paper, the definition of "difficulty" is open and task-specific. It is easy to find similar standards in other dataset. For VLN datasets like CVDN, each instance is organized as $<N_{0}, Q_{1}, A_{1}, N_{1}, \ldots, Q_{k}, A_{k}, N_{k}>$ where $N$ is navigation action, $Q$ is the question asked by the *Navigator*, $A$ is the answer from the *Oracle*. Thus, $k$ , the number of dialogue rounds, can partially represent the difficulty of the navigation task. Besides, there are many kinds of dialogue phenomena presented in [1] and each phenomena indicates a situation faced by the navigator. The situation has different complexity, hence can be used to evaluate the task. For Touchdown dataset, the task requires the agent to make right decision at each crossroad and find the Touchdown at the stop location. Therefore, the difficulty comes from both navigation and object detection. One can combine the number of crossroads the agent needs to pass-by and the number of attempts to find the Touchdown (mentioned in [2], should be available) to determine the task difficulty. Theses are prior knowledge based task difficulty decision. Besides, [3] discussed two difficulty decision methods, or equivalently, scoring methods, including transfer scoring and self-taught scoring. Transfer scoring means to recruit agents trained on other tasks and use the transferable knowledge to classify the samples. Self-taught scoring is to train the network using uniformly sampled mini-batches in current dataset and use the network output, like confidence score or loss value, to decide the task difficulty. The goodness of curriculum is to start from simple and gradually increase the task level. Hence, we think the criteria is not very important but the idea behind it matters.
>
>
>
> [1] Vision-and-Dialog Navigation
>
> [2] TOUCHDOWN: Natural Language Navigation and Spatial Reasoning in Visual Street Environments
>
> [3] On The Power of Curriculum Learning in Training Deep Networks

---

> > ### Comment · Reviewer_uKaH · 2021-08-23
> > **Response to author rebuttal**
> >
> > The author response is appreciated. The additional experiments conducted for Q2 are highly appreciated and the results answer my question -- results do indicate there is a clear benefit to using the prescribed curriculum in the right order. My concerns about limited contributions in this paper still hold and are echoed by other reviewers too -- (1) CL has been previously shown to work in the VLN domain [Zhu et al] (2) The SPCL methodology is same as [Jiang et al] (3) Experiments are conducted on only indoor VLN datasets. Overall, I am increasing my score from 5 to 6 given all the reviews and authors' response to all of them.

---

### Official Review · Reviewer_5Cjr · 2021-07-18

**Rating:** 6
**Confidence:** 4

**Summary:**

This paper introduces a new curriculum learning scheme for vision-and-language navigation (more particularly, the R2R dataset), where the navigation agent learns from easier to harder paths based on the number of traversed rooms. The curriculum learning scheme improves the performance of multiple baseline models on the R2R dataset.

**Limitations And Societal Impact:**

Yes.

**Main Review:**

This paper introduces a new curriculum learning scheme for vision-and-language navigation (more particularly, the R2R dataset), where the navigation agent learns from easier to harder paths based on the number of traversed rooms. The curriculum learning scheme improves the performance of multiple baseline models on the R2R dataset.

Strengths:
- The proposed curriculum learning (CL) scheme is intuitive and effective.
- It brings improvements to multiple baseline models on the R2R dataset.
- The paper is easy to read.

Weaknesses:
- Curriculum learning is not as novel as the authors claimed (certainly not the first CL method for VLN). [1] proposed a CL method for VLN already and [2] is also closely related. More discussions and comparison should be included, and the authors need to avoid overclaiming.
- The baseline navigation models used in the paper are sort of out-dated. How about some recent advanced VLN models such as VLN-BERT and Transformers? Would the CL work for them or not?
- Given that there are multiple datasets for VLN, new methods are typically expected to be validated on them. Particularly, the authors could have easily tested the CL approach on R4R. Why not?
- The CLS metric is not used in the paper.
- Regarding the ablation study on transfer learning in Table 5, [3] did some analysis on R2R and CVDN and showed that a simple yet effective multitask learning method can improve both datasets/tasks, which I think the authors should discuss.

[1] BabyWalk: Going Farther in Vision-and-Language Navigation by Taking Baby Steps
[2] Sub-Instruction Aware Vision-and-Language Navigation
[3] Environment-agnostic Multitask Learning for Natural Language Grounded Navigation




**Time Spent Reviewing:**

2

---

> ### Author Response · Authors · 2021-08-09
> **Reply to Reviewer 5Cjr**
>
> Thanks for all the constructive comments. Here is our reply to your questions.
>
> Q1:  Curriculum learning is not as novel as the authors claimed (certainly not the first CL method for VLN). [1] proposed a CL method for VLN already and [2] is also closely related. More discussions and comparison should be included, and the authors need to avoid overclaiming.
>
> A1:  For [1], we have discussed already in paper (Section 2, line 98 ~ 104). For [2], yes, sub-instruction and sub-paths can be considered simpler navigation tasks and hence they are naturally suitable to be included in a CL framework. We have did some experiments, that is, we combined both R2R and FGR2R dataset and train an agent using naive CL method on that. Results show improvement on both seen and unseen environments. We will conduct supplementary experiments and make some discussions in appendix. Our method is model-agonistic and can enhance the model performance without structure change, hence we call it the "first curriculum training paradigm", that is, it is not a model but a training way.
>
>
>
> Q2:  The baseline navigation models used in the paper are sort of out-dated. How about some recent advanced VLN models such as VLN-BERT and Transformers? Would the CL work for them or not?
>
> A2:  Thank you for reminding us the work based on transformers. For VLN-BERT, this work aims to use web-scraped vision-and-language materials to learn visual groundings that is transferable to VLN tasks. As stated in the paper, the training of VLN-BERT contains a generalized curriculum learning process, i.e. from language-only data, to web image-text pairs and finally to path-instruction pairs from the VLN dataset. Since our work focuses more on the curriculum inside path-instruction pairs from the VLN dataset, we did not make a direct comparison. We believe that using VLN-BERT as the backbone and adopting our method for fine-tuning can improve the result. The additional training data used by VLN-BERT does not conflict with our method (which does not need extra data). We think the agent's performance can benefit from both. We will supplement experiments in the next version.
>
>
>
> Q3:  Given that there are multiple datasets for VLN, new methods are typically expected to be validated on them. Particularly, the authors could have easily tested the CL approach on R4R. Why not?
>
> A3:  As you see, in our paper (Table 5) we tested CL method on both R2R dataset and RxR dataset. The reason why we did not tested the CL approach on R4R is that R4R is simply a modified version of R2R, it does not contain more annotations and is lack of language variety. For RxR dataset, data are collected by satisfying four path desiderata. Hence, it has different route distribution compared with R2R dataset and contains richer language phenomenon. Also, the scale of RxR dataset is larger. So we choose RxR dataset as an extension.
>
>
>
> Q4:  The CLS metric is not used in the paper.
>
> A4:  As we have discussed in our paper, we basically follow the standard metrics employed by (Speaker Follower, Fried et al. 2018). We add SPL and nDTW to better measure the agent's performance. Actually we have also computed SDTW value but did not present the results in paper due to the limit of page size. CLS is a metric that measures the fidelity of the agent’s path to the reference, whereas nDTW and SDTW represents the spatio-temporal similarity of the paths between the agent and the reference and covers more details. Both of these metrics are to measure the agent’s ability to follow instructions. As mentioned in [1], SDTW corresponds to human preferences the most. For a quick look, the CLS and SDTW for follower trained by normal method, naive cl method and spcl method are:
> \* on validation seen split, (CLS)   58.2%, 55.9% and 63.3%,  (SDTW)  44.1%, 40.5% and 50.5%
> \* on validation unseen split, (CLS)   35.0%, 34.4% and 38.9%,  (SDTW)  20.8%, 22.1% and 24.5%
> respectively. We are glad to add more details of both CLS and SDTW metric in the next version of our paper.
>
>
>
> Q5:  Regarding the ablation study on transfer learning in Table 5, [3] did some analysis on R2R and CVDN and showed that a simple yet effective multitask learning method can improve both datasets/tasks, which I think the authors should discuss.
>
> A5:  Yes, [3] proposed an effective multitask learning framework for both VLN and NDH task. The core idea of this work is to grasp common knowledge from another task/dataset so as to better the agent's performance on current task. Therefore, the experimental setup of [3] is significantly different with us, e.g. the model structure and the training process. Besides, our method is to improve the agent's performance within the scope of VLN task, even within a specific dataset, whereas [3] tries to build an adaptable navigation agent among multi-tasks. The range of information that the agent is based on between [3] and us differs. Hence, the experimental outcomes of [3] are not directly comparable with us. We will add the discussion in paper.
>
>
>
> [1] BabyWalk: Going Farther in Vision-and-Language Navigation by Taking Baby Steps
>
> [2] Sub-Instruction Aware Vision-and-Language Navigation
>
> [3] Environment-agnostic Multitask Learning for Natural Language Grounded Navigation

---

### Official Review · Reviewer_EhTe · 2021-07-20

**Rating:** 7
**Confidence:** 4

**Summary:**

This paper proposes a curriculum-based training paradigm for Vision-and-Language Navigation tasks. They focus on the aspect of whether the performance of a VLN agent can be improved without model structure change and data modification. They define the difficulty of the episode based on how many rooms it needs to cross and use this to divide the dataset into 5 parts. These parts are then used to form a curriculum learning training schedule. They use Self-Paced Curriculum Learning, and show that their method improves both the navigation performance and the training efficiency for the VLN task (R2R).

**Limitations And Societal Impact:**

They have explained limitations in the conclusions section.

**Main Review:**

Originality and Significance:
-	This is the first paper to develop a model-agnostic curriculum learning approach for VLN.
-	Their training curriculum improves the model performance without introducing any new data or increasing the model complexity.
-	In one of the experiments, they show that the curriculum learning paradigm helps the agent to improve the performance on the harder dataset without degrading the performance on the easier dataset.

Quality and Clarity:
-	The paper is well-written and easy to follow.
-	They have shown that their curriculum-learning approach consistently improves performance for 3 VLN-based models, Follower from Speaker-Follower paper, Self-Monitoring agent, and EnvDrop agent.
-	They have detailed their training algorithm and provided the hyperparameters to reproduce its results.


Typos:

L67, L181: argumentation -> augmentation

Figure 2: Bottom-left image: Longue -> Lounge

L91: ans -> and

L256: hrader -> harder

Table 5, caption: fisrt -> first


Minor Comment:

In figure 1, all images seem to have a blue-ish tint. Based on my experience working with the Matterport3D images, it looks like the images were outputted in BGR instead of RGB, please do check.



**Time Spent Reviewing:**

8

---

> ### Author Response · Authors · 2021-08-09
> **Reply to Reviewer EhTe**
>
> Thanks for all the constructive comments. Here is our reply to your questions.
>
> Q1:  In figure 1, all images seem to have a blue-ish tint. Based on my experience working with the Matterport3D images, it looks like the images were outputted in BGR instead of RGB, please do check.
>
> A1:  We have checked the plotting process, and yes, the blue-ish images are caused by the BGR output format. Thank you for pointing out this. We will fix this minor problem in the next version of the paper.
>
> With regard to the typos, thank you for reminding us of that. We will quickly correct those minor mistakes.

---

### Decision · Program_Chairs · 2021-09-27

**Decision:**

Accept (Poster)

**Comment:**

This work introduces a simple and intuitive approach to VLN based on the compositional nature of instructions, focusing therefore on single room, before multiform, etc.  The approach is then tested on multiple baselines for both R2R and RxR and shows consistent improvements.  The results are based on validation performance (not unseen test) and focus on older models which are substantially weaker than the current SotA on these domains.  This raises a natural question about the applicability of this approach to more contemporary models.